# Klotho Modulates Pro-Fibrotic Activities in Human Atrial Fibroblasts through Inhibition of Phospholipase C Signaling and Suppression of Store-Operated Calcium Entry

**DOI:** 10.3390/biomedicines10071574

**Published:** 2022-07-01

**Authors:** Yuan Hung, Cheng-Chih Chung, Yao-Chang Chen, Yu-Hsun Kao, Wei-Shiang Lin, Shih-Ann Chen, Yi-Jen Chen

**Affiliations:** 1Graduate Institute of Medical Sciences, National Defense Medical Center, Taipei 11490, Taiwan; dr.hungyuan@gmail.com; 2Division of Cardiology, Department of Internal Medicine, Tri-Service General Hospital, National Defense Medical Center, Taipei 11490, Taiwan; 3Division of Cardiology, Department of Internal Medicine, School of Medicine, College of Medicine, Taipei Medical University, Taipei 11031, Taiwan; michaelchung110@gmail.com; 4Division of Cardiovascular Medicine, Department of Internal Medicine, Wan Fang Hospital, Taipei Medical University, Taipei 11696, Taiwan; 5Department of Biomedical Engineering, National Defense Medical Center, Taipei 11490, Taiwan; yaochang.chen@gmail.com; 6Graduate Institute of Clinical Medicine, College of Medicine, Taipei Medical University, Taipei 11031, Taiwan; yuhsunkao@gmail.com; 7Department of Medical Education and Research, Wan Fang Hospital, Taipei Medical University, Taipei 11696, Taiwan; 8Faculty of Medicine and Institute of Clinical Medicine, National Yang-Ming Chiao Tung University, Taipei 112304, Taiwan; epsachen@ms41.hinet.net; 9Cardiovascular Center, Taichung Veterans General Hospital, Taichung 40705, Taiwan

**Keywords:** Klotho, atrial fibrillation, fibroblast, transient receptor potential channels

## Abstract

Background: Atrial fibroblasts activation causes atrial fibrosis, which is one major pathophysiological contributor to atrial fibrillation (AF) genesis. Klotho is a pleiotropic protein with remarkable cardiovascular effects, including anti-inflammatory, anti-oxidative, and anti-apoptotic effects. This study investigated whether Klotho can modulate the activity of human atrial fibroblasts and provides an anti-fibrotic effect. Methods: Cell migration assay and proliferation assay were used to investigate fibrogenesis activities in single human atrial fibroblasts with or without treatment of Klotho (10 and 100 pM, 48 h). Calcium fluorescence imaging, the whole-cell patch-clamp, and Western blotting were performed in human atrial fibroblasts treated with and without Klotho (100 pM, 48 h) to evaluate the store-operated calcium entry (SOCE), transient receptor potential (TRP) currents, and downstream signaling. Results: High dose of Klotho (100 pM, 48 h) significantly reduced the migration of human atrial fibroblasts without alternating their proliferation; in addition, treatment of Klotho (100 pM, 48 h) also decreased SOCE and TRP currents. In the presence of BI-749327 (a selective canonical TRP 6 channel inhibitor, 1 μM, 48 h), Klotho (100 pM, 48 h) could not inhibit fibroblast migration nor suppress the TRP currents. Klotho-treated fibroblasts (100 pM, 48 h) had lower phosphorylated phospholipase C (PLC) (p-PLCβ3 Ser537) expression than the control. The PLC inhibitor, U73122 (1 μM, 48 h), reduced the migration, decreased SOCE and TRP currents, and lowered p-PLCβ3 in atrial fibroblasts, similar to Klotho. In the presence of the U73122 (1 μM, 48 h), Klotho (100 pM, 48 h) could not further modulate the migration and collagen synthesis nor suppress the TRP currents in human atrial fibroblasts. Conclusions: Klotho inhibited pro-fibrotic activities and SOCE by inhibiting the PLC signaling and suppressing the TRP currents, which may provide a novel insight into atrial fibrosis and arrhythmogenesis.

## 1. Introduction

Atrial fibrillation (AF) is the most common clinical arrhythmia that raises the risk of major adverse cardiovascular events and has become a growing public health problem [1]. The pathophysiologic mechanisms of AF include ion channel dysfunction, abnormal calcium (Ca^2+^) homeostasis, structural remodeling, and autonomic neural dysregulation. Atrial fibrosis plays a significant role in both electrical and structural remodeling. Cardiac fibroblasts, which account for about 75% of total cardiac cells, can synthesize extracellular matrix (ECM) and modulate electrical characteristics of cardiomyocytes while they are coupling to cardiomyocytes [2]. These modulatory effects of fibroblasts coupling to cardiomyocytes include changes in pace-making function that generate abnormal spontaneous impulse formation and alternation of electrical conduction, which causes conduction barriers and slow conduction, which promote re-entry circuits [3]. 

Klotho, a type 1 single-pass transmembrane protein containing a large extracellular domain, is present in many organs, including the heart [4,5]. Soluble Klotho, the extracellular domain of Klotho cleaved and secreted into the blood, has multiple pleiotropic functions, including anti-oxidative, anti-inflammatory, and anti-apoptotic effects [5]. Haplodeficiency of Klotho gene increased collagen-1 expression and induced arterial stiffening [6]. Treatment of soluble Klotho suppressed myofibroblast proliferation and collagen production in cardiac myofibroblasts [7]. Chen et al. also reported that restoring serum soluble Klotho levels effectively suppressed inflammation and myofibroblastic transition in the aortic valve [8]. Several studies demonstrated that Klotho regulates transient receptor potential (TRP) channels to protect the cardiovascular system and maintain its integrity [5,9,10]. Additionally, Klotho was reported to suppress pulmonary and renal fibrosis [11,12,13]. However, knowledge of the role of Klotho in atrial fibrosis is limited. 

TRP channels, consisting of 6-transmembrane polypeptide subunits that assemble as tetramers forming cation-permeable pores, are extensively expressed in the heart and are responsible for Ca^2+^-signaling in both cardiac fibroblasts [14,15,16]. There are six subfamilies of the TRP channel family, including TRPC (canonical), TRPM (melastatin), TRPV (vanilloid), TRPA (ankyrin), TRPP (polycystin), and TRPML (mucolipin) families [15]. Several studies demonstrated that TRP channels are associated with human cardiac health and disease [17]. Among these TRP subgroups, TRPC was extensively studied in the regulation of cardiac function, and previous studies demonstrated that TRPC6 is responsible for myofibroblast trans-differentiation and inhibition of TRPC6 ameliorates tissue fibrosis [18,19]. Xie et al. also reported that Klotho could protect against stress-induced cardiac remodeling by blocking exocytosis of TRPC6 channels [10]. However, it is not clear whether Klotho may modulate the activity of TRP channels, leading to its anti-fibrosis potential. Accordingly, this study aims to evaluate whether Klotho modulates atrial fibroblasts activity and study the underlying mechanisms.

## 2. Materials and Methods

### 2.1. Cell Cultures

The NHCF-A human atrial fibroblasts cell line was purchased from Lonza Research Laboratory (Morrisville, NC, USA), and all fibroblasts were seeded as monolayers on uncoated culture dishes containing Dulbecco’s modified Eagle’s medium (Thermo Fisher Scientific, Loughborough, UK), including 100 U/mL penicillin-streptomycin (Thermo Fisher Scientific, Loughborough, UK) and 10% fetal bovine serum (HyClone Laboratories, Logan, UT, USA), in a humidified atmosphere containing 5% CO_2_ at 37 °C as previously described [20]. Human atrial fibroblasts used in this study were obtained from 4–6 passages to avoid the potential variations in the cellular structure and function.

### 2.2. Cell Migration Assay

A wound-healing assay was used to analyze the migration of human atrial fibroblasts treated with or without Klotho in different concentrations (10 pM and 100 pM, 48 h, R&D Systems, Minneapolis, MN, USA) 6 h after a cell monolayer in a six-well plate was craped with a P200 pipette tip. To explore the role of TRPC6 and intracellular signaling underlying the effects of Klotho in fibroblasts activity, we evaluated the migration in control, Klotho 100 pM-treated, BI-749327 1 μM (selective TRPC6 inhibitor, MedChemExpress, Monmouth Junction, NJ, USA)-treated, combined Klotho and BI-749327-treated, U73122 1 μM (phospholipase C inhibitor, Abcam, Cambridge, UK)-treated and combined Klotho and U73122-treated human atrial fibroblasts for 48 h. Each gap area was assessed using the ImageJ software (National Institutes of Health, Bethesda, Maryland, USA). The net migration areas after 6 h were subtracted from that at the time of the initial scratches.

### 2.3. Cell Proliferation Assay

As previously described, we used a commercial MTS kit (Promega, Madison, WI, USA) to measure the proliferation of human atrial fibroblasts [21]. Human atrial fibroblasts were seeded onto a 96-well culture dish at a density of 3000 cells per well. After growing to 50% confluence, the fibroblasts were incubated in a serum-free medium with Klotho in different concentrations (10 pM and 100 pM) for 48 h. Cell growth was analyzed by the MTS reagent, which was added 4 h before spectrophotometric analyses were performed. 

### 2.4. Patch-Clamp Experiments

We used the whole-cell patch-clamp technique to measure the ionic currents in human atrial fibroblasts with and without the administration of Klotho (100 pM, 48 h), BI-749327 (1 μM, 48 h), or U73122 (1 μM, 48 h) by using an Axopatch 1D amplifier (Axon Instruments, San Jose, CA, USA) at 35 ± 1 °C. The tip resistance of borosilicate glass electrodes containing pipette solution used in this study was 3–5 MΩ.

As described previously, the TRP current in human atrial fibroblasts was measured in the voltage-clamp configuration [22]. For recording TRP currents, the internal solution contained 120 mM CsCl, 9.4 mM NaCl, 1 mM MgCl_2_, 0.2 mM Na_3_GTP, buffered at 100 nM free Ca^2+^ with 10 mM BAPTA, and 10 mM HEPES (titrated with CsOH to pH 7.2), and the external solution contained 140 mM NaCl, 5 mM CsCl, 2 mM CaCl_2_, 1 mM MgCl_2_, 10 mM glucose, and 10 mM HEPES (titrated with NaOH to pH 7.4). We depolarized the human atrial fibroblasts by using triangular voltage ramps from –120 mV to +120 mV with a 200 ms pulse duration (from a holding potential of –60 mV with a frequency of 1 Hz). Considering TRP to be a non-selective cation channel, peak inward (-110 mV) and peak outward (+110 mV) current density were used to analyze the current amplitude.

### 2.5. Measurement of Intracellular Ca^2+^ Imaging

The human atrial fibroblasts were loaded on 1 cm glass-bottom chamber slides with the Fura-2-acetoxymethyl ester (5 μmol/L, Life Technologies, Carlsbad, CA, USA) and Pluronic F-127 (20% solution in dimethyl sulfoxide; 2.5 μg/mL) in a Ca^2+^-free solution containing 120 mM NaCl, 5.4 mM KCl, 1.2 mM KH_2_PO_4_, 1.2 mM MgSO_4_, 10 mM glucose, 6 mM HEPES, and 8 mM taurine (PH 7.40) for 40 min at 36 °C. We used a Polychrome V monochromator (Till Photonics, Munich, Germany) mounted on an upright Leica DMI 3000B microscope (Leica Microsystems, Buffalo Grove, IL, USA) with dual-excitation wavelengths of 340 and 380 nm and an emission wavelength of 510 nm to acquire Fura-2 fluorescence images. The MetaFuor software version 7.7.6.0 (Molecular Devices, Sunnyvale, CA, USA) was used to analyze Fura-2 images and the relative level of intracellular Ca^2+^ was represented by the ratio of fluorescence due to excitation at 340 nm (F340) to 380 nm (F380). To measure Ca^2+^ entry, all fibroblasts were initially superfused in the Ca^2+^-free solution for 2 min, following thapsigargin (2.5 μM) application. After thapsigargin induced Ca^2+^ release from the endoplasmic reticulum (ER), the extracellular solution was changed from a Ca^2+^-free solution to a solution containing 2mmol/L Ca^2+^ to measure Ca^2+^ entry via store-operated channels activated by ER Ca^2+^-store depletion. The Ca^2+^ entry was represented by the change in intracellular Ca^2+^ from a Ca^2+^-free solution to a 2 mmol/L Ca^2+^ solution (ΔF340/F380). 

### 2.6. Western Blot Analysis

Human atrial fibroblasts with and without the administration of Klotho (100 pM, 48 h) or U73122 (1 μM, 48 h) were homogenized and centrifuged in the buffer. We separated proteins electrophoretically on a polyacrylamide gel electrophoresis with 4% to 12% sodium dodecyl sulfate and electrophoretically transferred them to a polyvinylidene difluoride membrane that was equilibrated with sodium dodecyl sulfate as described previously [21]. We probed all blots with primary antibodies against procollagen type IA1 (Santa Cruz Biotechnology, Santa Cruz, CA, USA), procollagen type III (Abcam, Cambridge, UK), α-smooth muscle actin (α-SMA) (Abcam, Cambridge, UK), TRPC6 (Alomone Labs, Jerusalem, Israel), human PLC beta 3 (PLCβ3) (R&D Systems, Minneapolis, MN, USA), phospho-PLC beta 3 (Ser537) (pPLCβ3 Ser 537) (Cell Signaling Technology, Beverly, MA, USA); secondary antibodies were all conjugated with horseradish peroxidase. All bound antibodies were detected with an enhanced chemiluminescence detection system and analyzed using AlphaEaseFC^TM^ software (Genetic Technologies, Miami, FL, USA). All targeted bands were normalized to the glyceraldehyde 3-phosphate dehydrogenase protein (GAPDH) (Sigma-Aldrich, Merck, Darmstadt, Germany) to confirm equal protein loading.

### 2.7. Statistical Analysis

Means and standard errors of the means were used to describe all continuous variables. We used an unpaired *t*-test, Mann–Whitney rank-sum test, or one-way analysis of variance (repeated measures or non-repeated measures) with Tukey’s post hoc test to compare human cardiac fibroblasts under different treatment conditions. Statistical significance was defined as a *p*-value < 0.05. Statistics analyses were carried out using SPSS Statistic 18.0 software (Chicago, IL, USA) and SigmaPlot 12 software (Systat Software Inc, San Jose, CA, USA).

## 3. Results

### 3.1. The Effects of Klotho on the Migration, Collagen Production, Myofibroblast Differentiation, and Proliferation of Human Atrial Fibroblasts

Compared with control human atrial fibroblasts, a high dose of Klotho (100 pM)-treated atrial fibroblasts exhibited lesser migratory capability and lower expression of procollagen type IA1 and procollagen type III (Figure 1A,B). However, control and Klotho (100 pM)-treated atrial fibroblasts had similar expression of α-SMA. In contrast, control and Klotho-treated (10 and 100 pM) human atrial fibroblasts had similar proliferation rates (Figure 1C).

### 3.2. Klotho Reduced the Store-Operated Ca^2+^ Entry (SOCE), TRP Currents, and Phosphorylation of PLC in Human Atrial Fibroblasts

Since store-operated Ca^2+^ entry (SOCE) plays an essential role in the cellular calcium signaling and modulates cell migration [23], we measured the intracellular calcium imaging and compared the difference in SOCE between fibroblasts with and without treatment of Klotho. Klotho-treated (100 pM) human atrial fibroblasts exhibited a lower ER calcium release and a lower store-operated Ca^2+^ entry (SOCE) than control fibroblasts (Figure 2). Since TRP channels critically control the Ca^2+^ influx and regulate the cardiac fibroblast function [24], we examined whether Klotho may modulate the TRP current. In this experiment, Klotho-treated (100 pM) human atrial fibroblasts had lower TRP currents than control fibroblasts (Figure 3A). Because previous studies demonstrated that Klotho inhibited the PLC signaling [25] and PLC modulates TRP channels and SOCE [26,27,28], we also performed Western blots to examine the effect of Klotho on PLC signaling. As shown in Figure 3B, Klotho-treated (100 pM) human atrial fibroblasts had similar expression of total PLCβ3 but less expression of pPLCβ3 Ser537 than control fibroblasts. TRPC6 channels have been shown to modulate fibroblast function and promote fibrosis [18,29]. Accordingly, we evaluated the role of TRPC6 in human atrial fibroblasts and found that BI-749327 (a selective TRPC6 inhibitor, 1 μM)-treated fibroblasts had smaller TRP currents than control fibroblasts. However, in the presence of Klotho (100 pM), fibroblasts with and without treatment of BI-749327 had similar TRP currents, which were smaller than control fibroblasts (Figure 3A). Moreover, Western blots showed that Klotho (100 pM)-treated and control fibroblasts had a similar expression of TRPC6 (Figure 3B). These findings suggest that Klotho might modulate TRPC6 function, leading to reduced atrial fibroblasts migration. We performed the migratory capability test of fibroblasts and found that Klotho (100 pM)-treated fibroblasts, BI-749327 (1 μM)-treated fibroblasts, and combined Klotho (100 pM) and BI-749327 (1 μM)-treated fibroblasts had lower migratory capability than control fibroblasts with a similar extent (Figure 3C,D).

### 3.3. The PLC Inhibitor Suppressed the Fibroblasts Migration, Collagen Production, SOCE, and TRP Currents of Human Atrial Fibroblasts

As shown in Figure 4A, Klotho (100 pM)-treated fibroblasts, PLC inhibitor (U73122, 1 μM)-treated fibroblasts, and combined Klotho (100 pM) and PLC inhibitor (1 μM)-treated fibroblasts had lower migratory capability than control fibroblasts similarly. Western blots showed that Klotho (100 pM)-treated fibroblasts, PLC inhibitor (U73122, 1 μM)-treated fibroblasts, and combined Klotho (100 pM) and PLC inhibitor (1 μM)-treated fibroblasts had lower expression of procollagen IA1, procollagen III, and pPLCβ3 Ser 537 than the control. However, the expressions of α-SMA and total PLCβ3 were similar in the four groups (Figure 4B).

Klotho (100 pM)-treated fibroblasts, PLC inhibitor (U73122, 1 μM)-treated fibroblasts, and combined Klotho (100 pM) and PLC inhibitor (1 μM)-treated fibroblasts had a lower ER release and a lower SOCE than control fibroblasts with a similar extent (Figure 5). Moreover, The Klotho (100 pM)-treated fibroblasts, PLC inhibitor (U73122, 1 μM)-treated fibroblasts, and combined Klotho (100 pM) and PLC inhibitor (1 μM)-treated fibroblasts had similarly lower TRP peak inward currents and lower TRP peak outward currents than control fibroblasts (Figure 6A–C). However, Western blots showed that fibroblasts in the four groups had similar expressions of TRPC6 (Figure 6D).

## 4. Discussion

### 4.1. Klotho Attenuates Pro-Fibrotic Effects in Human Atrial Fibroblasts

AF is the most common arrhythmia in clinical practice and contributes to cardiovascular adverse events such as heart failure and ischemic stroke [1]. The essential mechanisms of AF genesis include contractile, structural, and electrical remodeling of atrial tissues, and atrial fibrosis plays a crucial role in pathophysiological remodeling [30]. Atrial fibroblasts, one major component of atrial fibrosis, secrete various ECM, including procollagen type I and type III, and differentiate into myofibroblasts in response to pathological stimuli [31]. During atrial fibrosis, disorganized interstitial fibrosis changes the longitudinal conduction and the electronic coupling of myofibroblasts and cardiomyocytes, facilitating re-entrant arrhythmia [32]. In addition, fibrosis decelerates action potential propagation and enhances ectopic automaticity, contributing to arrhythmogenesis [3,33]. Klotho is a type 1 single-pass transmembrane protein with pleiotropic functions, including attenuating pro-fibrotic response [8,11,12,13,34,35,36]. Initially, Klotho was found to ameliorate renal fibrosis [37]; however, emerging evidence suggests that Klotho attenuates fibrosis in extra-renal tissues, such as pulmonary and cardiovascular systems [8,13]. In the present study, Klotho inhibited atrial fibroblasts migration in a dose-dependent manner and reduced procollagen synthesis without altering fibroblasts proliferation. These findings suggest that Klotho could suppress atrial fibrosis and potentially decrease fibrosis-associated conduction disturbance in atrial cardiomyocytes. Additionally, Klotho may be involved in another mechanism of protection from AF. Klotho can upregulate the expression of heat shock proteins, which protect from atrial tachycardia-induced remodeling and AF perpetuation [38,39]. The similarity in the α-SMA expression between control and Klotho-treated fibroblasts suggests that Klotho might not modulate the fibroblast activity via the regulation of myofibroblast differentiation. However, the cell culture environment with hard plastic dishes used in this study may not correlate well to the in vivo environment that may enhance myofibroblast differentiation with significant expression of α-SMA. Therefore, it is still not clear whether Klotho may potentially modulate fibroblast differentiation.

### 4.2. Klotho Modulates TRP Currents and Calcium Handling in Atrial Fibroblasts

Previous studies demonstrated that cytosolic Ca^2+^ is essential for the regulation of the activation of fibroblasts [40,41,42], and our previous study showed that Klotho modulates electrical activities and intracellular Ca^2+^ in pulmonary vein cardiomyocytes [43]. The present study revealed that Klotho (100 pM) significantly suppressed the thapsigargin-induced ER Ca^2+^ release and the subsequent Ca^2+^ entry. These findings exhibited an association between inhibition of intracellular Ca^2+^ through SOCE and suppression of fibroblast functions, such as collagen synthesis and migration, in fibroblasts treated with Klotho (100 pM).

Intracellular signaling of cardiac fibroblasts that govern fibroblast behavior is complex, among which Ca^2+^ signaling plays a vital role in cardiac fibrosis, including the synthesis of extracellular matrix proteins and migration [42]. Recently, SOCE and non-selective cation channels have been found to be critical to fibroblast Ca^2+^ signaling. Among numerous non-selective cation channels regulating the Ca^2+^ entry and subsequent extracellular signal-related kinase pathways [16,24,44], TRP channels activate atrial fibroblasts by modulating Ca^2+^ signaling and mediate fibrogenesis in AF [6,44]. TRPCs, one class of TRP channels, have been postulated to form both receptor-operated Ca^2+^ entry and SOCE [45]. Nikolova-Krstevski reported that TRPC6 channels expressed on the atrial endocardium modulate myocardial Ca^2+^ regulation in response to mechanical stretch, which contributes to the structure and electrical remodeling [46]. PLC signaling, critical in fibroblast activation, results in the generation of inositol 1,4,5-triphosphate (IP_3_) and diacylglycerol (DAG). Consequently, DAG activates protein kinase C, which promotes atrial fibrosis through modulating fibroblast function and stimulating collagen synthesis, and IP_3_ activates the IP_3_ receptor and induces rapid release of the luminal Ca^2+^ from the ER [47]. Depletion of ER Ca^2+^ activates the stromal interaction molecule that subsequently triggers SOCE through promoting Ca^2+^ release-activated Ca^2+^ channel protein 1, forming a complex with TRPCs, contributing to extracellular Ca^2+^ influx [48,49]. In addition, PLC-mediated DAG generation could activate TRPC6 and enables Ca^2+^ influx [50,51]. In our experiment, Klotho (100 pM) suppressed the TRP peak inward currents and peak outward currents significantly; however, the additional inhibitory effect was not observed in a selective TRPC6 inhibitor. These findings might imply that the inhibition of TRPC6 currents contributed to the main inhibitory effect of Klotho on the TRP current without changing the expression of TRPC6. Additionally, Klotho (100 pM) suppressed the pPLCβ3 (Ser537) expression while the total PLCβ3 expression remained unchanged. Our findings suggest that Klotho might modulate TRPC6 function by inhibiting PLC signaling.

### 4.3. Klotho Regulates the Intracellular Signaling in Atrial Fibroblasts

Klotho is a coreceptor of fibroblast growth factor (FGF) receptors, and it triggers signal propagation of FGFs, such as FGF19, FGF21, and FGF23 [52]. Loss of cardiac Klotho expression facilitates transforming growth factor-β1 pathway and aggravates cardiac fibrosis [53]. Klotho has also been shown to modulate intracellular calcium signaling and regulate ion channels on the plasma membrane, including Na^+^/K^+^-ATPase, TRPV5, TRPC6, and renal outer medullary potassium channel 1 [5,54]. A previous study demonstrated that soluble Klotho inhibited FGF23-stimulated PLC and phosphoinositide 3-kinase/Akt signaling [25]. In the present study, as compared to the control, Klotho (100 pM)-treated fibroblasts, U73122 (1 μM)-treated fibroblasts, and combined Klotho (100 pM) and U73122 (1 μM)-treated fibroblasts had lower fibroblasts migration, less procollagen IA1/III synthesis, and less phosphorylation of PLCβ3 (Ser537) with a similar extent. In addition, Klotho (100 pM)-treated fibroblasts, U73122 (1 μM)-treated fibroblasts, and combined Klotho (100 pM) and U73122 (1 μM)-treated fibroblasts had reduced thapsigargin-induced ER Ca^2+^ release and the subsequent extracellular Ca^2+^ entry to a similar extent. The above findings suggest no synergetic inhibitory effects of Klotho and U73122 on atrial fibroblasts. Thus, Klotho might exhibit its inhibitory effects on fibroblast migration, collagen synthesis, and SOCE by inhibiting the PLC signaling. Figure 7 summarizes the hypothesized mechanisms underlying the anti-fibrosis effects of Klotho. 

### 4.4. Limitation

First, although we found that Klotho modulates PLC signaling and subsequent SOCE, the detailed mechanisms underlying Klotho’s regulation in human atrial fibroblasts are not fully elucidated. More experiments are mandatory to clarify the modulatory effects of Klotho on IP_3_R and following stromal interaction molecule activation. Second, human atrial fibroblasts from passages 4–6 were used in this study, and their fibroblast characteristics might be different from those in patients. Third, several conditions such as aging and heart failure could enhance cardiac fibrosis and AF genesis. Future experiments from pathological animal models are useful to confirm Klotho’s potential in reducing atrial fibrosis and AF prevention.

## 5. Conclusions

Klotho modulates the pro-fibrotic activities and intracellular calcium homeostasis in human atrial fibroblasts through its inhibitory effect on the PLC pathway and decreases SOCE by suppressing TRPC6 currents, which may provide a novel therapeutic insight into atrial fibrosis and AF genesis.

## 6. Highlights

Klotho decreased migration and expressions of procollagen type IA1 and procollagen type III in human atrial fibroblasts.Klotho reduced TRP currents, mainly due to the decreased TRPC6 currents, and subsequently modulated fibroblasts activity.Klotho inhibited the PLC pathway and decreased thapsigargin-induced ER Ca^2+^ release and SOCE in human atrial fibroblasts.

## Figures and Tables

**Figure 1 biomedicines-10-01574-f001:**
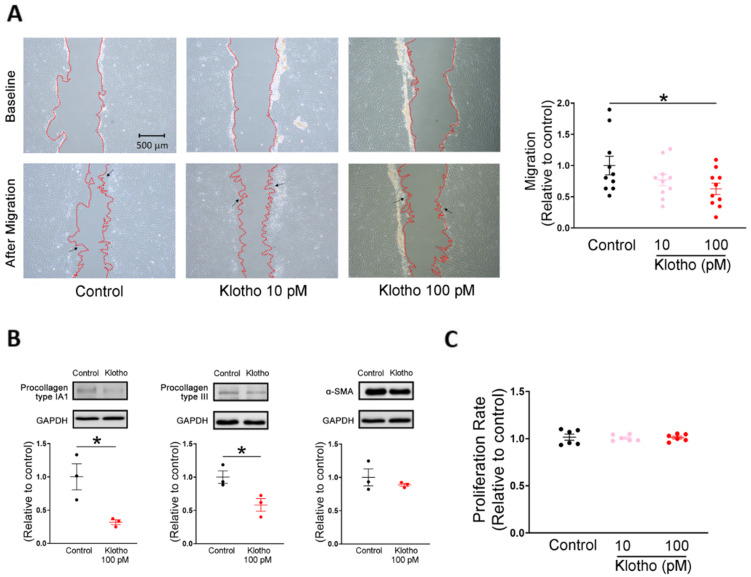
Cell migration, collagen production, and proliferation capabilities in human atrial fibroblasts treated with and without Klotho. (**A**) Example photographs and average data (*n* = 10 from 3 independent experiments) showed that human atrial fibroblasts treated with Klotho at a high concentration (100 pM) exhibited less migratory capability than the control. Arrows indicate migrating fibroblasts in the wound space. (**B**) Example photographs and average data revealed that lower procollagen type IA1 expression (*n* = 3 from 3 independent experiments) and lower procollagen type III expression (*n* = 3 from 3 independent experiments) in human atrial fibroblasts treated with a high concentration of Klotho (100 pM). The expression of α-smooth muscle actin (α-SMA) was comparable in both groups (*n* = 3 from 3 independent experiments). GAPDH was used as a loading control. (**C**) Treatment with both lower and high concentrations of Klotho (10 and 100 pM) for 48 h had no significant effect on the proliferation rate of human atrial fibroblasts. (*n* = 6 from 3 independent experiments). GAPDH, glyceraldehyde 3-phosphate dehydrogenase protein. * *p* < 0.05.

**Figure 2 biomedicines-10-01574-f002:**
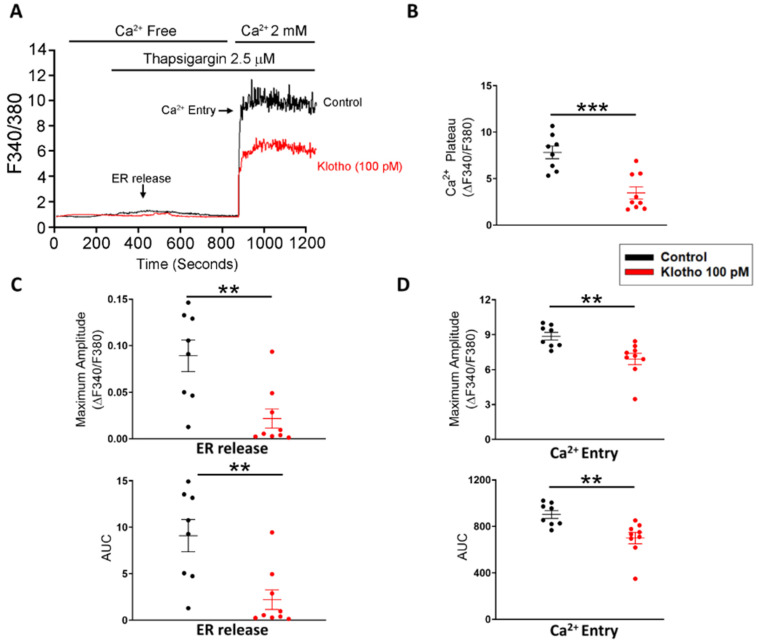
Store-operated Ca^2+^ entry (SOCE) in human atrial fibroblasts with and without Klotho (100 pM). Representative tracings (**A**), average data of the Ca^2+^ plateaus (**B**), the endoplasmic reticulum (ER) release (**C**) and Ca^2+^ entry (**D**) in Fura-2 AM-loaded human atrial fibroblasts. Klotho (100 pM, 48 h)-treated fibroblasts (*n* = 8–9 from 4 independent experiments) had a lower ER release and a lower Ca^2+^ entry than control fibroblasts (*n* = 8 from 3 independent experiments). ER, endoplasmic reticulum. ** *p* < 0.01, *** *p* < 0.005.

**Figure 3 biomedicines-10-01574-f003:**
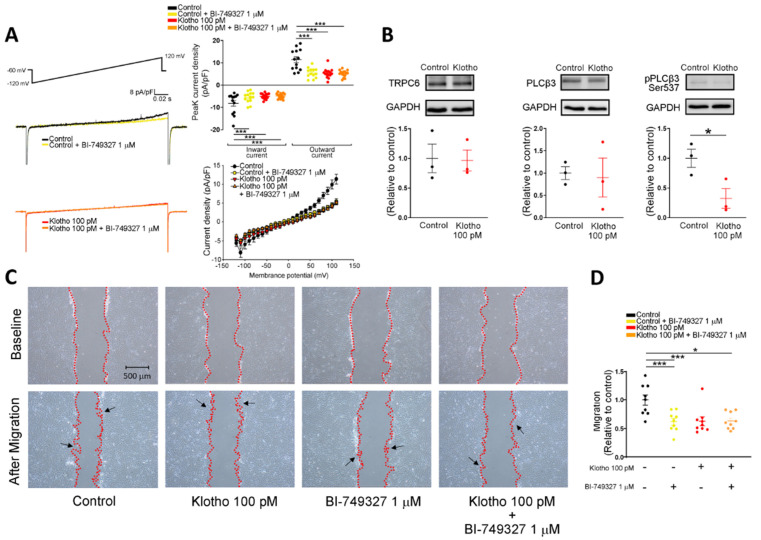
Effects of Klotho (100 pM) and selective TRPC6 inhibitor (BI-749327, 1 μM) on ionic currents of transient receptor potential canonical channel (I_TRPC_) in human atrial fibroblasts. (**A**) The current tracings and I–V relationship of I_TRP_ showed that BI-749327-treated fibroblasts (1 μM, 48 h, *n* = 13–15 from 4 independent experiments), Klotho-treated fibroblasts (100 pM, 48 h, *n* = 15 from 3 independent experiments), and combined Klotho and BI-749327 (*n* = 15 from 3 independent experiments) had similarly lower peak inward currents and similarly lower peak outward currents compared with the control (*n* = 13 from 4 independent experiments). (**B**) Western blots showed a similar expression of TRPC6 and total phospholipase C (PLC) between fibroblasts treated with Klotho (100 pM, 48 h) and control fibroblasts. However, the phosphorylated PLC (p-PLC) expression was lower in the fibroblasts treated with Klotho (100 pM, 48 h) (*n* = 3 from 3 independent experiments). (**C**,**D**) Example photographs and average data (*n* = 9 from 3 independent experiments) showed that human atrial fibroblasts treated with BI-749327 (1 μM, 48 h), Klotho (100 pM, 48 h), and combined Klotho and BI-749327 exhibited similarly less migratory capability than the control. Arrows indicate migrating fibroblasts in the wound space. GAPDH was used as a loading control. PLCβ3, phospholipase C beta 3; pPLCβ3, phosphorylated phospholipase C beta 3; TRPC6, transient receptor potential canonical channel 6. * *p* < 0.05, *** *p* < 0.005.

**Figure 4 biomedicines-10-01574-f004:**
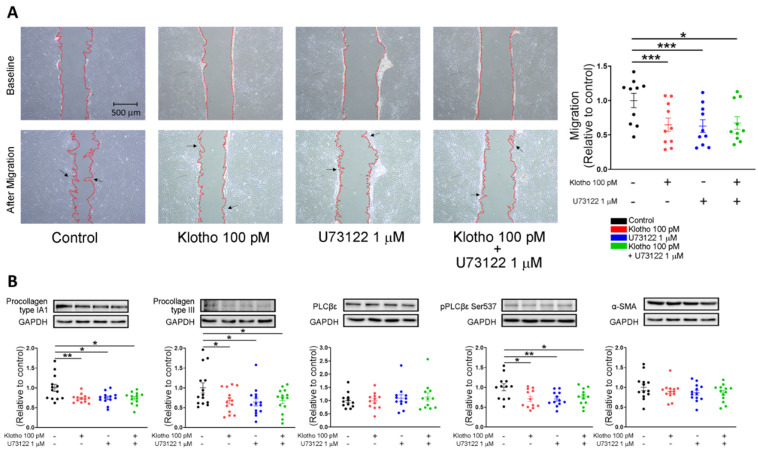
Cell migration and collagen production in human atrial fibroblasts in 4 different groups (control; Klotho 100 pM, 48 h; U73122 1 μM, 48 h; Klotho 100 pM plus U73122 1 μM, 48 h). (**A**) Example photographs and average data (*n* = 9 from 3 independent experiments) showed the cell migration of human atrial fibroblasts in 4 different groups. Arrows indicate migrating fibroblasts in the wound space. (**B**) Example photographs and average data revealed the expression of procollagen type IA1, procollagen type III, total PLC, p-PLC, and α-SMA in human atrial fibroblasts in 4 groups (*n* = 12–14 from 3 independent experiments). GAPDH was used as a loading control. * *p* < 0.05, ** *p* < 0.01, *** *p* < 0.005.

**Figure 5 biomedicines-10-01574-f005:**
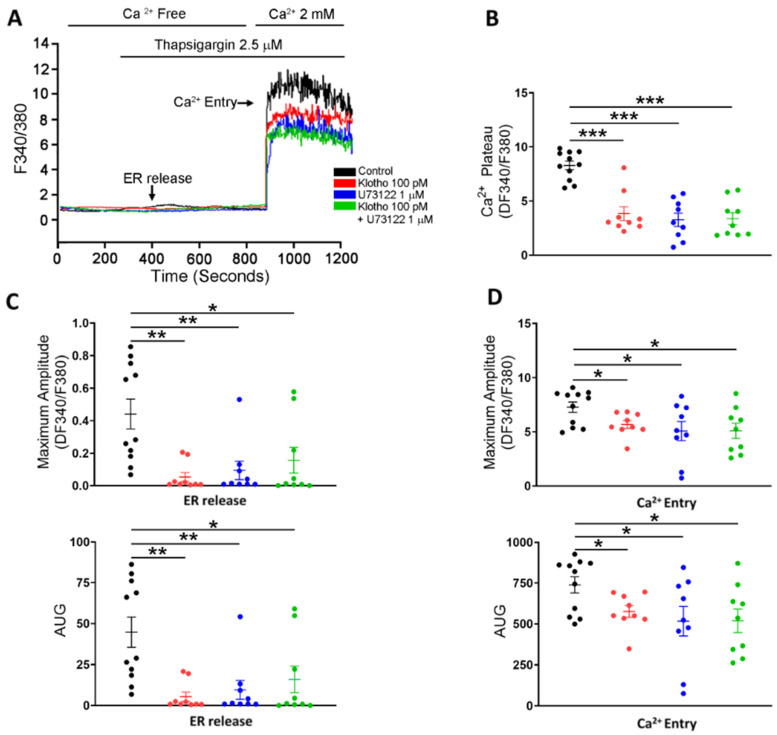
SOCE in human atrial fibroblasts in 4 different groups (control; Klotho 100 pM, 48 h; U73122 1 μM, 48 h; Klotho 100 pM plus U73122 1 μM, 48 h). Representative tracings (**A**) and average data of the Ca^2+^ plateaus (**B**), the ER release (**C**), and the maximum amplitude of the Ca^2+^ entry (**D**) in Fura-2 AM-loaded human atrial fibroblasts (*n* = 9–11 from 4 independent experiments). Human atrial fibroblasts in the control group had a higher ER release and a higher Ca^2+^ entry than the other 3 groups. * *p* < 0.05, ** *p* < 0.01, *** *p* < 0.005.

**Figure 6 biomedicines-10-01574-f006:**
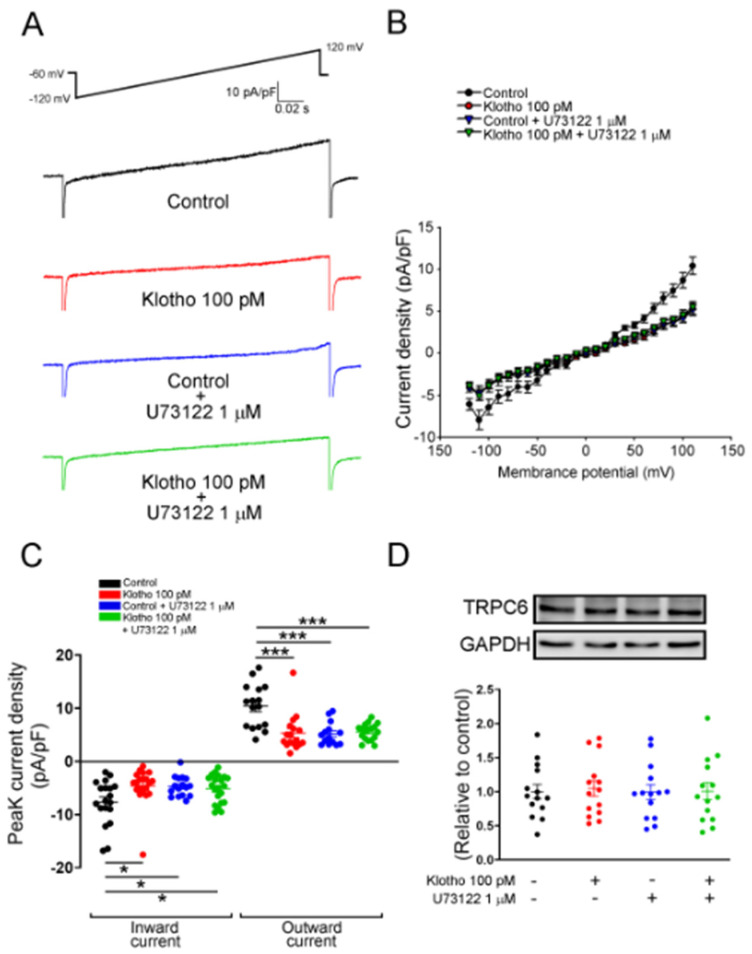
I_TRP_ of human atrial fibroblasts in 4 different groups (control; Klotho 100 pM, 48 h; U73122 1 μM, 48 h; Klotho 100 pM plus U73122 1 μM, 48 h). The current tracings (**A**) and I–V relationship (**B**) of I_TRPC6_ of human atrial fibroblasts in 4 groups (*n* = 16 from 5 independent experiments). (**C**) Fibroblasts in the control group had higher peak inward currents and outward currents than the other 3 groups (*n* = 14–18 from 5–7 independent experiments). (**D**) Western blots showed similar expression of TRPC6 among the four groups. (*n* = 14 from 3 independent experiments). GAPDH was used as a loading control. * *p* < 0.05, *** *p* < 0.005.

**Figure 7 biomedicines-10-01574-f007:**
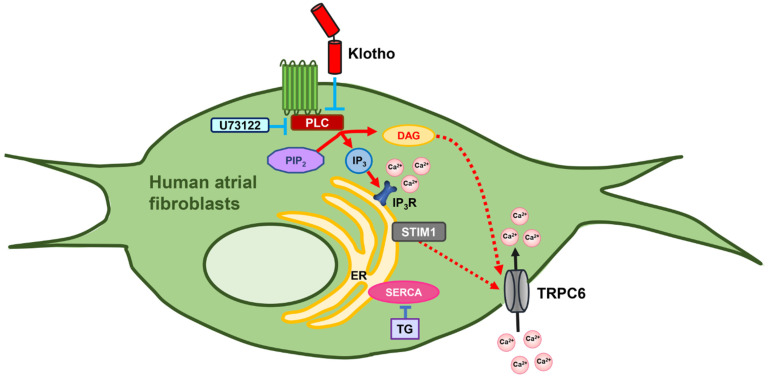
Proposed mechanism of action of Klotho in the modulation of ion channels and intracellular calcium handling in human atrial fibroblasts. Klotho inhibits the activation of the PLC-IP_3_ signaling and suppresses thapsigargin-induced ER Ca^2+^ release, and subsequently attenuates SOCE by inhibition of TRPC6 currents. In addition, Klotho may also inhibit the PLC-DAG pathway and negatively modulate the inward currents of TRPC6. PIP_2_, phosphatidylinositol biphosphate; PLC, phospholipase C; DAG, diacylglycerol; IP_3_, inositol 1,4,5-triphosphate; IP_3_R, IP_3_ receptor; ER, endoplasmic reticulum; STIM1, stromal interaction molecule 1; SERCA, sarcoendoplasmic reticulum calcium transport ATPase; TG, thapsigargin.

## Data Availability

All data generated or analyzed during the current study are included in this published article and are available from the corresponding author on reasonable request.

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
