# Peer review of "Klotho Modulates Pro-Fibrotic Activities in Human Atrial Fibroblasts through Inhibition of Phospholipase C Signaling and Suppression of Store-Operated Calcium Entry"

_biomedicines, 2022, doi:10.3390/biomedicines10071574_

Round 1

Reviewer 1 Report

Title:  Klotho Modulates Pro-fibrotic Activities in Human Atrial Fibroblasts through Inhibition of Phospholipase C Signaling and Suppression of Store-Operated Calcium Entry Authors Yuan Hung, Cheng-Chih Chung, Yao-Chang Chen, Yu-Hsun Kao, Wei-Shiang Lin, Shih-Ann Chen, Yi-Jen Chen   COMMENTS:  The submitted manuscript describes an interesting study in terms of the important medical problem. The Authors have revealed the inhibitory effects of Klotho on pro-fibrotic activities of human atrial fibroblasts and elucidated the molecular mechanism of the revealed phenomena that is based on modulation of calcium entry. This material can be helpful for readers of Biomedicines and may be publishes. There are only few points of minor criticism:  1.  The pictures of restoring cell monolayer in Fig. 1A and Fig. 4A are not very convincing. The inhibitory effects of Klotho are not clearly seen in those Figs. Maybe, it would be better to present the bars only (?).  2. In page 11, line 341, the Authors used the term 'tumor growth factor-beta'. Probably, they mean TGF-beta - transforming growth factor-beta (?). If yes, it should be corrected from 'tumor' to 'transforming'.  3. It would be nice, if in Discussion, the Authors mention that Klotho may be involved in another mechanism of protection from atrial fibrillation: e.g. Klotho promotes upregulation of Hsp70 (DOI: 10.3892/etm.2021.9917),  while Hsp70 can preserve from atrial fibrillation ( DOI: 10.1161/01.RES.0000252323.83137.fe  ).       4. It is unclear from the Authors' conclusions: whether Klotho may be used as a therapeutic tool against atrial fibrillation?           

Reviewer 2 Report

This is an interesting paper addressing an important issue on the role of klotho in atrial fibrillation. Other recent publications have already linked klotho with cardiovascular and kidney diseases (10.1093/ndt/gfy126; 10.1177/2040622320940176)

The present study shed more light on the possible explanation of how klotho may exert its effect on human fibroblasts. The study is well conductec and experimental protocol well designed.

However, how this protein might be useful in clinical practice, need to be better addressed with well designed clinical trials

Reviewer 3 Report

The paper titled : << Klotho Modulates Pro-fibrotic Activities in Human Atrial Fibroblasts through Inhibition of Phospholipase C Signaling and 3 Suppression of Store-Operated Calcium Entry >> is authored by Hung Y et al.

The authors evaluated the role of Klotho, a pleiotropic protein with anti-inflammatory effects, on cardiac fibroblasts’ proliferation.

 This study of fibroblasts (FB) is well conducted and the authors provide original results on the role of Klotho on FB proliferation.

Some concerns could be address to strengthen the message:

1. Transform all bar graphs into scatter/dot-plot graphs showing mean and error bars.

2. In scratch wound assay’s figures, please use arrows to indicate a fibroblast in the wound space.

3. Scale is unreadable in wound assay figures please improve.

4. Add a highlight section with up to 5-bullet points to summarize the main findings of the paper.

4. Please organise the discussion in headings to better clarify your demonstration

5. Add a distinguishable Limitation section to describe what could habe been done to consolidate your study, and to assess possible future directions.
